# Koi (*Cyprinus rubrofuscus*) Seek Out Tactile Interaction with Humans: General Patterns and Individual Differences

**DOI:** 10.3390/ani11030706

**Published:** 2021-03-05

**Authors:** Isabel Fife-Cook, Becca Franks

**Affiliations:** Department of Environmental Studies, New York University, New York, NY 10003, USA; ifc219@nyu.edu

**Keywords:** human–animal interaction, human–animal relationship, *Cyprinus rubrofuscus*, koi, carp, aquatic veterinary science, ornamental fish, captive animal welfare, fish welfare, animal protection, empathy, positive welfare, environmental enrichment, personality, cognition

## Abstract

**Simple Summary:**

To assess the possibility of forming stable and trusting relationships between humans and fishes, we documented the interactions between a familiar human experimenter and seven koi (*Cyprinus rubrofuscus*). Analyses of video data shows that overall, koi spent more time than expected in close proximity to the human and even sought out physical contact. Moreover, individual fish displayed unique interaction patterns, with some frequently engaging in tactile interaction and others only periodically or rarely doing so. By demonstrating that koi will voluntarily interact with humans and that individual differences play an important role in interaction style, this study provides the first evidence that individuated human–fish relationships may be possible, which has powerful implications for how we think about, treat, protect, and provide care for fish.

**Abstract:**

The study of human–animal interactions has provided insights into the welfare of many species. To date, however, research has largely focused on human relationships with captive mammals, with relatively little exploration of interactions between humans and other vertebrates, despite non-mammals constituting the vast majority of animals currently living under human management. With this study, we aimed to address this gap in knowledge by investigating human–fish interactions at a community garden/aquaponics learning-center that is home to approximately 150 goldfish (*Carassius auratus*) and seven adult and two juvenile koi (*Cyprinus rubrofuscus*). After a habituation period (July–September 2019) during which time the fish were regularly provided with the opportunity to engage with the researcher’s submerged hand, but were not forced to interact with the researcher, we collected video data on 10 non-consecutive study days during the month of October. This procedure produced 18~20-min interaction sessions, 10 during T1 (when the experimenter first arrived and the fish had not been fed) and eight during T2 (20–30 min after the fish had been fed to satiation; two sessions of which were lost due equipment malfunction). Interactions between the researcher and the seven adult koi were coded from video based on location (within reach, on the periphery, or out of reach from the researcher) and instances of physical, tactile interaction. Analyses revealed that overall, koi spent more time than expected within reach of the researcher during both T1 (*p* < 0.02) and T2 (*p* < 0.03). There were also substantial differences between individuals’ overall propensity for being within-reach and engaging in physical interaction. These results show that koi will voluntarily interact with humans and that individual koi display unique and consistent patterns of interaction. By providing quantitative data to support anecdotal claims that such relationships exist around the world, this research contributes to the ongoing discoveries highlighting the profound dissonance between how humans think about and treat fish and who fish actually are, thereby emphasizing the necessity of stronger moral and legal protections for fishes.

## 1. Introduction

Human–animal interaction (HAI) is an umbrella term used to describe any form of interaction, relationship or bond between a human and nonhuman [1]. HAI research is inherently multidisciplinary, consisting of a diverse body of literature from many academic fields including anthropology, biology, psychology, sociology, ecology, ethology and veterinary medicine with applications across a wide range of contexts [2]. Historically, HAI research has focused primarily on the relationship between companion animal ownership and human health, the therapeutic effects of structured human–animal interactions (e.g., Animal Assisted Interventions), and the consequences of poor stockmanship on farmed animal production [1]. More recently, HAI literature has seen a marked increase in contextual diversity, with a growing body of work focused on impacts of HAI on animal welfare [3] as well as the conception of an interdisciplinary literature exploring the mutual benefits of human–animal bonds and relationships [3,4], a development consistent with the mounting interest in positive animal welfare within the broader scope of animal welfare science [5,6,7]. Despite this progress, the species diversity of HAI research has remained relatively low, maintaining a focus on mammals and some birds to the exclusion of the vast majority of animals who do not fall under these classifications.

One such understudied taxonomic group is the Osteichthyes, a diverse class of aquatic vertebrates colloquially known as “bony fishes” or simply “fish”. Osteichthyes are used by humans for a wide variety of reasons (including food, recreation, ornamentation, scientific research, and companionship) and on an enormous scale. According to the Food and Agriculture Organization (FAO), an estimated 0.98 and 2.85 trillion individuals are killed for human consumption each year, making fish the most consumed taxa on the planet [8]. Zebrafish (*Danio rerio*) are popular model organisms increasingly used in studying vertebrate development and gene function in laboratories around the world [9,10] and freshwater fish are currently the most populous pet in the US, with an estimated 139.5 million individuals currently living in homes across the country [11]. Though historically perceived as “lesser vertebrates”, recent research has established that fish are not only sentient but capable of sophisticated emotional and cognitive processes [12,13], emphasizing the ethical significance of researching, legislating and reinforcing welfare standards for fishes living in captivity. Despite this necessity for fish welfare research, it was not until recently that scientists began to investigate fish welfare systematically [14]. Even now, however, the existing fish research remains largely focused on mitigating negative consequences of captivity with a paucity of literature exploring psychological and social welfare of fishes [15].

Amongst the many families of Osteichthyes used by humans, cyprinids (commonly referred to as carps) are by far the most populous both in number and diversity of use [16]. Cyprinids are the largest family of vertebrates with over 1300 extant species [17], many of which are farmed and fished for food (e.g., common carp (*Cyprinus carpio*)), used as model organisms in biomedical research (e.g., zebrafish (*Danio rerio*)), and bred for ornamental purposes (e.g., “fancy carp” such as goldfish (*Carassius auratus*) and koi (*Cyprinus rubrofuscus;* correctly referred to as Nishikigoi but henceforth referred to simply as “koi”)). Koi originated in East Asia and, after thousands of years of selective breeding for striking coloration patterns, have become a widely popular ornamental species [18,19]. Today, koi are widely available in most countries and remain popular additions to water gardens and home aquaria around the world. Yet, like other fish, no research to date has investigated human–koi interactions.

Human–animal interactions can impact the welfare of captive animals in a multitude of ways. Nonhuman animals can suffer as the result of negative human contact (e.g., inflicting pain [20,21,22,23], inducing fear and anxiety [24,25,26,27]) or poor management (e.g., failing to provide adequate cognitive and emotional stimulation in captivity leading to boredom and loneliness [28,29,30]). Nonhuman animals can also benefit from positive human contact that provides pleasant sensory stimulation [31,32] and promotes improved cognitive function through positive reinforcement training [31,32]. Experiencing or expecting positive HAI such as prosocial interactions can also lead humans to improve how they treat animals directly under their care [33,34]. Moreover, emotional bonds between humans and companion animals can have a profound positive effect on animal welfare, with humans providing comfort in stressful situations and offering social companionship [2,35,36]. Finally, HAIs can also affect animal welfare indirectly by influencing human perceptions and attitudes [37,38,39,40,41,42,43]. For example, affiliative HAIs such as engaging in playful behavior [44], interacting physically [45] and observing pro-social behaviors between conspecifics in naturalistic settings [46], can foster empathy in humans, which can then increase public support for stricter welfare standards and conservation efforts [47].

The growing body of evidence emphasizing the diversity of ways in which HAIs impact the welfare of nonhuman animals has led a number of influential ethologists to propose integrating HAI into animal welfare evaluations and monitoring schemes [48,49,50], including the recently updated five domains model of animal welfare [51]. These developments and discoveries make the absence of knowledge about HAIs for many important and heavily impacted species particularly concerning.

The present study aimed to approach this literature gap by examining the interactions or lack thereof between a familiar human experimenter and seven koi residing in a mixed-species pond at a community garden. The first step in determining what role HAIs may play in fish welfare and protection is establishing whether voluntary human–fish interactions are possible. We hypothesized that the possibility for positive human–fish interactions would be evidenced by fish voluntarily approaching and interacting with the human experimenter. Alternatively, if the fish were fearful of or indifferent to humans, we expected to find evidence of avoidance or random swimming patterns. We were also interested in determining whether individual differences (i.e., personalities) played a role in driving human–fish interactions, in which case, we expected to find stable patterns of interaction style across time. Alternatively, if human–fish interactions were mainly driven by external, environmental cues (e.g., fear of humans or general attraction to novelty/disturbance), we expected to find group-level patterns dominating their behavioral variability, with little to no difference in individual koi behavior. To test these hypotheses, we filmed human–fish interactions over the course of a month and coded the videos for (i) the fishes’ proximity to the researcher’s submerged hand and (ii) instances of fish engaging in voluntary tactile interaction with the researcher.

## 2. Materials and Methods

### 2.1. Location and Subjects

Data collection took place at Oko Farms, a small-scale community-managed aquaponics learning center located in Brooklyn, NY. Oko Farms is the permanent home of ~150 goldfish (*Carassius auratus*) and 9 koi (*Cyprinus rubrofuscus*). The fish varied in size, age, breed and background, with the goldfish primarily purchased from commercial pet stores or surrendered by citizens in groups or individually and the koi purchased from various retailers and private suppliers. The youngest fish living at Oko Farms were goldfish fry who were born in the spring of 2019. The oldest fish were 5 adult koi who were purchased between 5 months and 2 years of age and have lived on the farm since 2014. Thus the 7 koi in this study ranged from at least 2 years to at least 6 years of age. We focused on the behavior of the 7 adult koi (see Figure 1 and Table 1) because they were the most readily identifiable individuals in the tank and were present for the entire acclimation period (the two juvenile koi were not included in the study as they arrived after the onset of the acclimation period).

### 2.2. Animal Care

#### 2.2.1. Housing

All fish are housed in a below-ground tank protected by a greenhouse connected to outdoor grow beds. The total water volume of the flow-through aquaponics system is roughly 15,000 gallons with approximately 4000 gallons in the holding tank, 2500 of which are physically accessible to the fish who participated in this study. The holding tank is roughly 7.5 ft wide, 12 ft long and ranges between 3 and 4 feet deep. The tank is lined with pond-liner and barren apart from four aerators in each corner of the tank.

#### 2.2.2. Feeding

Fish were fed between 9 and 16 oz of commercial carp feed 2–3× per day by either the primary researcher or the farm operator. On study days, the fish were not fed for at least two hours prior to the primary interaction period, after which they were fed the prescribed amount according to the predetermined feedings schedule. Feed time and amount were recorded and updated daily. Feed intake decreased over the course of the study as a result of decreasing water temperatures due to seasonal changes.

#### 2.2.3. Maintenance

Staff members observe the fish and test water quality at least once per day. The water is tested using an API freshwater master test kit and the parameters (temperature, pH, total ammonia, nitrite and nitrate) are recorded on a daily basis. Over the course of the study, the parameters remained relatively stable aside from the temperature, which dropped in conjunction with seasonal changes. Ammonia ranged between 0 and 0.5 ppm, nitrite at 0 ppm and nitrate between 0 and 10 ppm. Temperature ranged between 70 and 60 F over the course of the study and pH ranged between 7 and 7.2.

### 2.3. Study Timeline

#### 2.3.1. Habituation Period (July 2019–October 2019)

The habituation period involved socializing the fish to the presence of humans for ~3 h per day, 5 days a week, resulting in a total of approximately 190 h of socialization over the course of 3.5 months. When entering the fishes’ environment, one of three researchers sat beside the tank with one or both hands submerged in the water up to the elbows. The tank is large enough that by sitting at the side of the tank, the fish could remain well away from our submerged hands while still allowing curious animals the opportunity to approach. Researchers strove to limit sudden movements or loud noises that may frighten the fish and kept their faces within sight of the fish whenever possible. It should be stressed that, during this period, researchers did not instigate physical contact with any of fish but rather allowed the fish to choose whether or not they were interested in interacting. If fish instigated physical interaction (e.g., touching, mouthing, etc.) the researchers returned contact (e.g., gently stroking the fish’s forehead, wiggling fingers, etc.).

Though at the onset of the study, all fishes were accustomed to the presence of humans outside their tank, the fishes were unaccustomed to the presence of a human hand in their environment for extended periods of time. Furthermore, the main experience the fishes had with a human entering their environment was of being netted, a process known to be aversive to fish [52,53,54,55]. In order to counteract the learned fear response to humans in their environment, we fed the fish by hand throughout the month of July and into early August, at which point the fish readily approached us on sight with no sign of aversion. At this point we ceased hand feeding but continued to interact normally while providing the fish access to our submerged hands throughout the remainder of the acclimation period.

#### 2.3.2. Study Period (4 Weeks between October and November 2019)

The data collection portion of the study was carried out on 9 days over the course of 4 weeks from 8 October to 5 November. Each observation period involved an approximately twenty-minute session during which the researcher (the same individual for the entire data collection portion) knelt next to the tank, submerged her hand in water and remained attentive to the fish. Sessions were filmed using an iPhone mounted on a light pole propped next to the lip of the tank.

### 2.4. Data Collection and Analysis

#### Video Data

Videos were filmed on an iPhone XR with a 12 MP wide-angle camera (A1984, Apple) attached to a phone mount adapter (Cell Phone Tripod Mount Adapter, PHONE-CLAMP, Fotodiox Inc., New York, NY, USA) screwed into an adjustable light stand (Heavy-Duty Light Stand Black 13′, LS-13HBI, Impact) positioned above the holding tank. At the end of each study session, videos were uploaded onto the researcher’s personal computer (MacBook Pro Retina, 13”, Early 2015, A1502, Apple), backed up on an external hard drive (WD 1TB Black My Passport for Mac Portable External Hard Drive, WDBJBS0010BSL-NESN, Western Digital, Kingston, NY, USA) and uploaded to a private YouTube account.

Behavioral analyses were performed by uploading videos into BORIS (Behavioral Observation Research Interactive Software, version 7.10.2) [56] which was used to extract data using the coding scheme outlined in the following section. Cumulative duration calculations were performed in BORIS and the resulting data downloaded as a spreadsheet. SketchUp 3d (version 20.0.362) modeling software was used to calculate the area of the tank and standardize the location data.

### 2.5. Coding Scheme

Videos were coded according to the animals’ relative proximity to the researcher (designated by the researcher’s estimated reach) and physical interactions with the researcher: within reach, periphery, and outer area (see Figure 2, Table 2).

### 2.6. Statistical Analyses

To assess overall patterns of behavior, we used multilevel modeling (also known as hierarchical models or mixed-effects models, which are a more flexible version of repeated measures ANOVA), to control for random effects of study day and repeated sampling of the same individual fish over time. This statistical approach is useful for accommodating unequal sample sizes while also correcting for pseudoreplication [57,58]. For hypothesis testing within the multilevel models, we used t-tests and Satterthwaite corrected degrees of freedom.

To determine whether fish spent more time than expected in each area (within-reach, periphery, vs. outer-area) we calculated an “area-adjusted duration” for each fish within each area for each observation. Area-adjusted duration is the percent of time a fish spent in a location during an observation period divided by that location’s percent of total area (% duration/% area). If a fish moved randomly through the tank, their duration percentages would be equal to the area percentages, leading to an area-adjusted duration of 1. If a fish spent more time in a certain location than would be expected from a random-movement model, the area-adjusted duration would be greater than 1 (and it would be less than 1 if they spent less time than expected in a certain location).

To test for overall patterns in location preference, we compared the area-adjusted durations for within-reach, periphery, vs. outer-area by using area-adjusted duration as the outcome variable and location as input variable (also known as the independent variable). We controlled for observation time period (T1 vs. T2) as a fixed-effect and study day and fish ID as random effects.

All calculations, analyses, and visualizations were performed in R statistical software [59,60]. For data cleaning and visualization, we used the tidyverse package [61]. For statistical modeling we used the lme4 and lmerTest packages [62,63].

## 3. Results

Overall, the fish spent significantly more time than expected within-reach than they did in the periphery: area-adjusted duration was 0.37 (standard error [SE] = 0.07) higher for within-reach than periphery (t(310.28) = 5.06, *p* < 0.0001). They spent significantly more time than expected in the periphery than they did in the outer-area: area-adjusted duration was 0.32 (SE = 0.07) higher in periphery than outer-area (t(310.28) = 4.32, *p* < 0.0001). Logically, therefore, within-reach was also higher than outer-area (t(310.28) = 9.38, *p* < 0.0001; Figure 3). Examining the within-reach patterns in more detail, we found that while fish spent more time within reach during T1 than T2 (t(316.90) = 2.52, *p* < 0.02), for both time periods, overall duration in the within-reach location was significantly higher than what would be expected from a random-movement model: T1 area-adjusted duration for the within-reach location was 1.75 (SE = 0.20; t(6) = 3.69, *p* < 0.02) and T2 area-adjusted duration for the within-reach location was 1.35 (SE: 0.11; t(6) = 3.11, *p* < 0.03; Figure 3).

Fish showed strong individual differences within these patterns, however. Margaret spent the most amount of time within-reach: 2.38 (SE = 0.40) times more than expected during T1 (t(9) = 3.49, *p* < 0.007) and 1.71 (SE = 0.31) times more than expected during T2 (t(7) = 2.26, *p* < 0.06). Gabriel spent the second-most amount of time within-reach: 2.22 (SE = 0.37) times more than expected during T1 (t(9) = 3.31, *p* < 0.01) and 1.67 (SE = 0.32) times more than expected during T2 (t(7) = 2.07, *p* < 0.08). The individual who spent the least amount of time within-reach, Tigerlily, spent an average of 0.95 (SE = 0.22) times less than expected during T1 (t(9) = 0.24, *p* > 0.8) and 0.95 (SE = 0.10) times less than expected in T2 (t(7) = 0.48, *p* > 0.6, Figure 4).

Fish also showed strong individual differences in their tendency to seek out physical, tactile interaction with the human experimenter. Gabriel’s rate of physical interaction was highest, seeking physical contact on average 0.63 (SE = 0.16) times per minute during T1 and 0.29 (SE = 0.09) times per minute during T2, whereas other fish rarely engaged in this behavior (see Figure 5 and Table 3 for a breakdown of physical interaction rates by fish).

The cumulative time fish spent in tactile contact with the experimenter also varied strongly by individual. Gabriel spent a median of 49.1 (range: 10.2, 281) seconds in physical contact with the human during T1 and a median of 18 (range: 0, 54.7) seconds during T2. While all study fish initiated in at least some physical interaction, several had median physical interaction durations of 0 (See Figure 6 and Table 4 for a breakdown of total physical interaction durations by fish).

## 4. Discussion

The results of this study demonstrate that captive carp can and do seek out physical contact with a familiar human and that they show individual differences in interaction patterns. Though there is a wealth of data on the benefits of positive human–animal relations, no previous work has explored their potential in a species of fish. The results presented here thus provide a foundation for future work investigating the characteristics and consequences of positive human–fish relations. The following sections present a number of hypotheses regarding the findings of this study and discuss various promising avenues for future research on human–fish interactions.

### 4.1. Reducing Stress through Socialization

Fear of humans can pose a significant threat to welfare for animals housed in perpetual proximity to humans [26]. Fear of humans can be quickly learned and retained for long periods of time [64]; for example, dairy calves who had been treated aversively quickly learned to recognize and avoid the perpetrator, eventually generalizing the fear to all handlers after repeated negative experiences [65]. Limiting human-induced fear and anxiety is paramount to captive animal welfare as it can easily lead to chronic stress when animals are unable to rectify or escape from the aversive stimuli [25]. Chronic stress, in turn, has been linked to poor health and behavioral abnormalities in a variety of captive animals [66,67], including fishes [68,69,70,71].

Establishing positive associations with human contact has been shown to decrease stress in a number of captive species across a variety of contexts [65,72,73], allowing for easier handling and safer veterinary procedures [74] for both human and nonhuman participants. The present results show that fish have the potential to form non-fear-based relationships with humans, which has implications for their welfare. First, the presence of such relationships may reduce overall stress compared to animals living with a fear-based relationship with humans. Second, trusting relationships with humans may mitigate the negative effects of stress associated with necessary handling and relocating animals, a recognized welfare concern for captive fishes [25,52,53,54,75,76,77]. Future work is required to explore these possibilities.

### 4.2. Human–Fish Interactions: A Potential Form of Enrichment for Captive Fishes?

The koi showed interest in pursuing physical interaction even after feeding to satiation, suggesting that they may be motivated to seek physical interaction for reasons unrelated to food seeking behavior.

#### 4.2.1. Pleasant Sensory Experience

One potential explanation for the fishes’ interest in physical contact is that interacting with a novel substance and texture (human skin) serves as a source of tactile and/or sensory enrichment. Sensory stimulation is an important aspect of welfare for many animals, including humans, for whom massage therapy reduces stress [78] by stimulating the release of endorphins (opiates produced by the brain that trigger feelings of relaxation) such as oxytocin and vasopressin [79]. Pleasant tactile interaction plays an important role in developing positive human–animal relationships and can influence animal welfare in a number of ways. For instance, exposure to pleasant human contact from a young age is a reliable method of decreasing fear and stress in many farmed animal species, including cows [80,81,82], sheep [83,84], chickens [85] and pigs [86,87,88]. Under certain circumstances, pleasant tactile stimulation can also serve as a source of sensory enrichment for captive animals [89], as supported by research on the effects of pleasant physical manipulation on horses [90] and dogs [91]. In both cases, the animals’ heart rates dropped when being groomed by humans, mirroring the physiological response elicited by engaging in allogrooming with conspecifics thus suggesting that tactile stimulation by humans may also be perceived as rewarding.

Experiencing pleasure as the result of physical interaction with a receptive conspecific partner is believed to be a fundamental and highly conserved element of both human and nonhuman welfare. While touch in fishes has not yet been explored from an explicitly welfare angle, there is biological and behavioral evidence to suggest that teleost fishes possess the physiological prerequisites necessary for experiencing pleasurable sensation [92]. Further research on sensory stimulation in fish may prove fruitful when designing enrichment strategies [89], a hypothesis supported by research on sensory seeking behavior in fish (e.g., [92,93,94]. For instance, Soares et al. [93] conducted a study on the effects of tactile stimulation on the surgeonfish (*Ctenochaetus striatus*), a tropical reef species that regularly visits cleaner wrasses (*Labroides dimidiatus*) to have ectoparasites removed. Previous studies showed that cleanerfish are able to influence client fish decisions by physically touching the surgeonfish with its pectoral and pelvic fins [94]. Soares et al. simulated this behavior by exposing surgeonfish to mechanically moving cleanerfish models, which resulted in significantly lower levels of cortisol in fish stimulated by moving models compared to those exposed only to stationary models. These results show that physical contact alone is enough to produce fitness-enhancing benefits. Furthermore, because the fish visited the model by their own volition, it can be reasonably assumed that the tactile stimulation of the cleanerfish model elicited a pleasurable reaction. This hypothesis is further supported by evidence that cleanerfish routinely use tactile “massage” as a consolation or reward for disgruntled clients in the wild, indicating that the client fish finds this sensation inherently pleasurable [94]. These studies not only support the notion that some fishes enjoy tactile stimulation but also suggest that interspecies interactions—perhaps including such as those between humans and fishes—may be rewarding as well.

#### 4.2.2. Cognitive and Social Enrichment

A complimentary hypothesis as to why the fish spent more time than expected within reach during both T1 and T2 is that their interest in interaction may be motivated by curiosity, suggesting that fish-directed human interaction may serve as a means of cognitive enrichment by presenting an opportunity to explore and exercise agency. Research on the effects of boredom on captive animal welfare suggest that boredom is likely experienced by a wide range of captive species and is generally perceived as aversive, particularly by those living in barren conditions and lacking the agency necessary to remedy the situation, and thereby leading to frustration and anxiety [28,30]. Cognitive stimulation is an effective tool in staving off boredom and thus is increasingly recognized as an essential component of animal welfare in captivity [28,95,96,97]. Human interaction can provide captive animals, particularly those living in relatively barren environments, with a degree of arousal that their living conditions fail to fulfill [40,98,99,100]. This work suggests the possibility that fish-directed human interaction could serve as a source of cognitive stimulation for some species of captive fishes. Future work is needed to explore this possibility.

### 4.3. Koi Personality, Sociality and Future Directions in Carp Welfare Research

Analyses of location and behavior data (see Figure 3, Figure 4, Figure 5 and Figure 6) reveal that domestic carp display distinct individual patterns of interaction, including duration of location behavior and individual propensity for tactile interaction with the researcher. Gabriel showed both the highest average rate and longest cumulative duration of physical interaction during both T1 and T2, followed by Maggie and Margaret. Bessie, Tigerlily and Gingko showed the lowest average rates of physical interaction (see Table 4 and Figure 6).

These results provide preliminary insight into the personalities of the seven fish who participated in the study, particularly in their interest in seeking interaction and willingness to take the risk of interacting. Interest in interacting, as defined by the relative amount of time spent within reach and interacting physically, ranged substantially between individuals. For example, the comparatively low rates of physical interaction and time spent within reach of the two least interactive fishes (Bessie and Tigerlily) signify a general lack of interest in interaction, but also did not show a substantial avoidance of the researcher. On the other hand, Gingko, whose rate and duration of physical interaction was similar to those of Bessie and Tigerlily, spent substantially more time within reach than the other two least interactive fishes, a behavior profile indicating attentiveness and interest but less interest in initiating physical interaction. Dominick showed stable interaction patterns across the board, with very little variation between T1 and T2 and a relatively low propensity to linger within reach of the researcher when not interacting.

Investigating these individual behavior profiles can benefit fish welfare in several ways. First, provided that the patterns are found to be relatively stable over time, they can serve as a baseline by which to evaluate individual welfare. Recognizing and tracking patterns of individual behavior is a useful tool in measuring welfare in zoo housed animals [101,102,103], with significant changes in individual and/or group behavioral dynamics considered to pose a potential welfare threat. While this approach is appropriate for certain conditions (such as poor water quality), it may not be sufficient in detecting subtle signs of early disease, many of which are critical to identify early in order to prevent major losses [104]. These community level patterns and associated affective states have been studied in other species of fish (e.g., [105,106].) but never in koi. The individual behavioral patterns observed in the study suggest that researchers, farmers and hobbyists may also be able to use individual personality traits and their associated behavior patterns to better refine welfare parameters for koi.

### 4.4. Human–Fish Interactions and Animal Protection—Promoting Human Empathy and Compassion through Positive Interspecies Exchanges

As discussed briefly in the introduction, it has been well-documented that interactions with non-human animals, when perceived as positive by the human participant, can have a positive effect on humans’ perception of the species in question [34]. Human perceptions and emotions influence actions and decisions and as such, improving human attitudes towards a species can lead to improvements in their care and strengthen support for their protection [47,107]. Given the relatively uncharitable public opinion towards fish in general and lack of awareness around fish cognition, sentience and emotion [108,109,110], research establishing the potential for positive human–fish interactions is particularly needed to counteract baseline expectations and improve perceptions of fishes’ worth and moral status. Additional work exploring human–fish interactions thus has the potential to have significant downstream consequences for the welfare of fishes including increasing public support for fish protection and regulations.

## 5. Conclusions

In this study, we investigated whether or not koi would voluntarily choose to interact with a familiar human and, if so, whether individual behavior patterns could be identified. We found that koi not only engaged with the human both before and after feeding but also that the fish remained within reaching distance of the human more often than expected based on a random motion model during all sessions. These results suggest that koi are not solely motivated to interact with a familiar human in anticipation of a food reward and that the baseline interest in interaction may be motivated by a desire to interact for its own sake. Additionally, we found substantial, individual patterns of interaction behavior and proclivity for interaction between the seven koi participating in the study, suggesting that voluntary HAI may be used to investigate animal personality in conjunction with traditional personality assessment paradigms. Overall, this study provides important preliminary evidence that human–fish interaction research will be a fruitful area of future inquiry. More work is needed in examining human-mediated enrichment opportunities for fish, determining the qualitative nature of the human–fish interactions, and exploring the implications of carp sociality and personality for their welfare in captivity.

## Figures and Tables

**Figure 1 animals-11-00706-f001:**
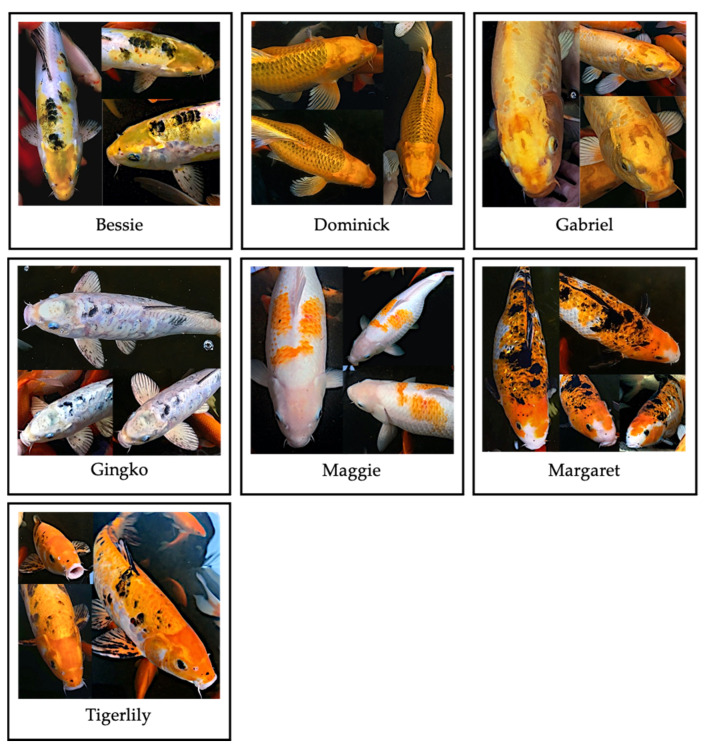
Participants. From left to right: Bessie, Dominic, Gabriel, Gingko, Maggie, Margaret, Tigerlily.

**Figure 2 animals-11-00706-f002:**
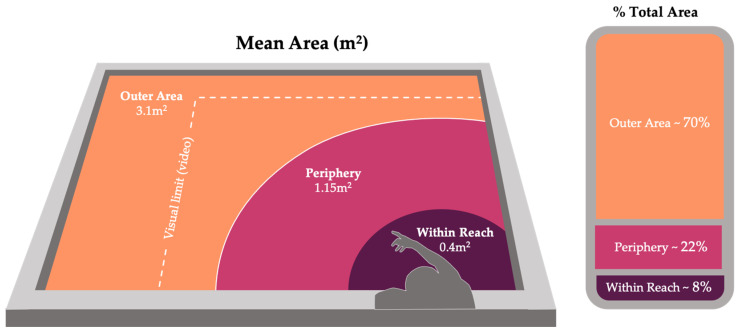
Visual representation of location codes used to standardize variable zone size (see Table 2). *Graphics by Isabel Fife-Cook*.

**Figure 3 animals-11-00706-f003:**
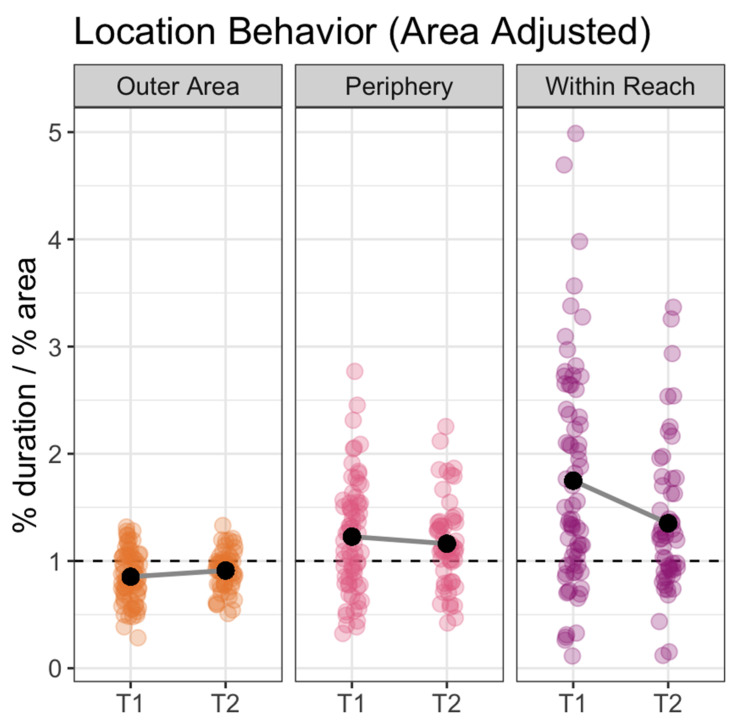
Plot showing area adjusted duration in each of the three locations: Outer Area, Periphery and Within Reach. During both T1 and T2, fish spent more time than expected within reach (*p* < 0.03). Each point represents the area-adjusted duration for one fish over the course of a single video. T1 = study sessions immediately following arrival and T2 = study sessions taken 20–30 min after feeding. The dotted line at 1 indicates baseline (expected) duration if movement is random. Plot points higher than 1 indicate that the fish spent more time than expected in the corresponding location during the course of the study session while plot points below 1 indicate that the fish spent less time than expected in the location.

**Figure 4 animals-11-00706-f004:**
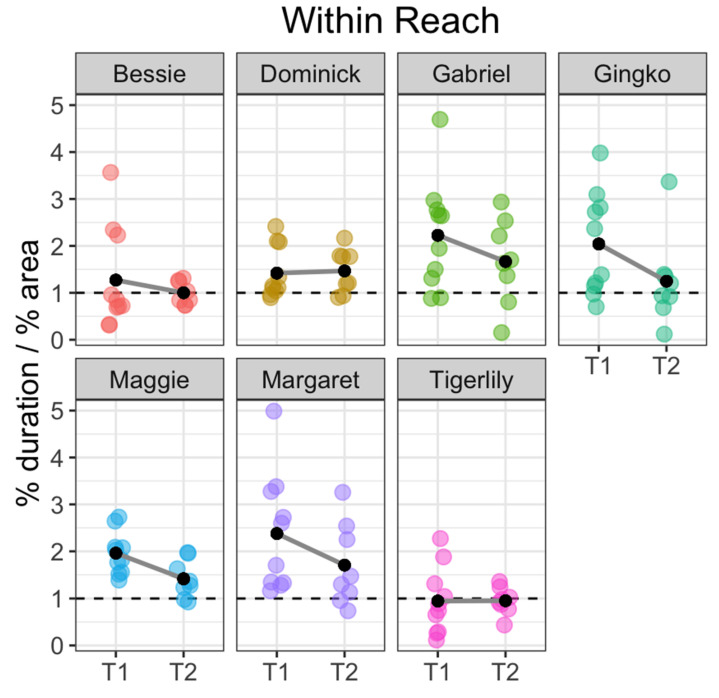
Plot showing area adjusted duration for the within-reach location by fish. Each colored point represents the area adjusted duration for each fish over the course of a single session. Black dots represent the mean area-adjusted duration for that fish during that session type: T1 = study sessions immediately following arrival and T2 = study sessions taken 20–30 min after feeding. The dotted line at 1 indicates baseline (expected) duration if movement is random. Plot points higher than 1 indicate that the fish spent more time than expected in the corresponding location during the course of the study session while plot points below 1 indicate that the fish spent less time than expected in the location.

**Figure 5 animals-11-00706-f005:**
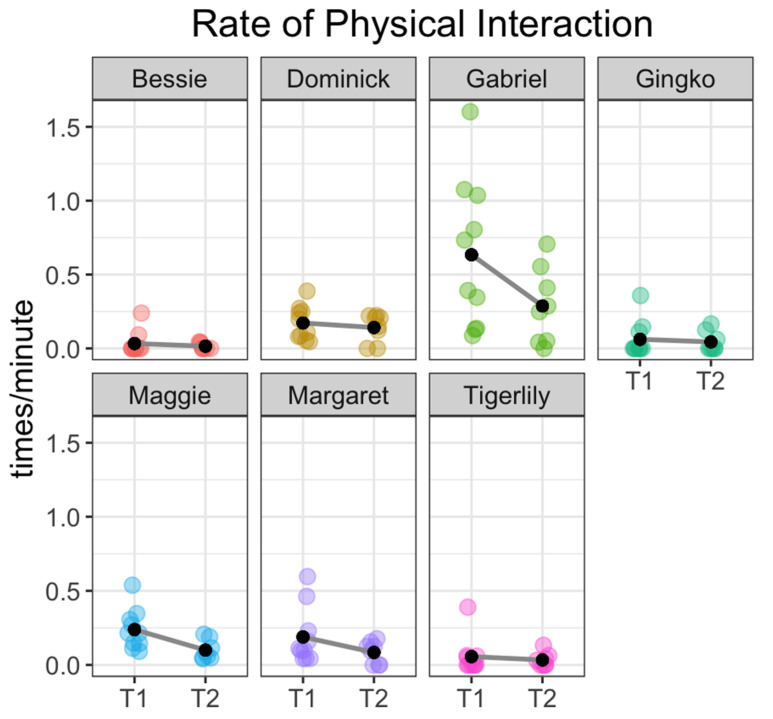
Plot showing the rate of physical interaction for each fish. Each colored point represents the rate of Physical Interaction for each fish during a single study session. Black dots represent the mean physical interaction rate for that fish during that session type: T1 = study sessions immediately following arrival and T2 = study sessions taken 20–30 min after feeding.

**Figure 6 animals-11-00706-f006:**
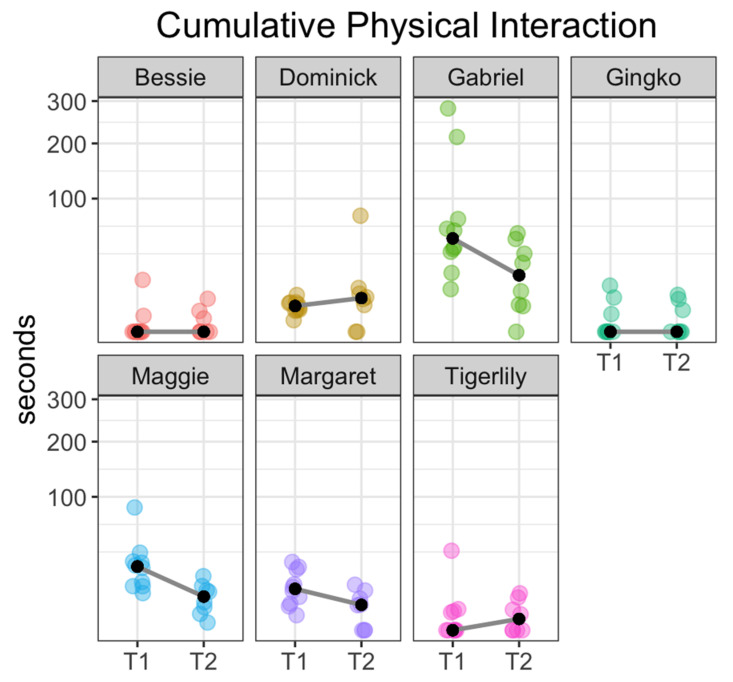
Plot showing cumulative duration of physical interaction for each fish. Each colored point represents the cumulative duration of physical interaction for the corresponding fish over the course of a single session. Black dots represent the median cumulative duration for that fish during that session type: T1 = study sessions immediately following arrival and T2 = study sessions taken 20–30 min after feeding. *Y*-axis is on a square-root scale to accommodate the skewed distribution.

**Table 1 animals-11-00706-t001:** Fish profiles.

Name	Sex	Length	Distinguishing Features
Bessie	M	40–45 cm	Primarily grey with lemon yellow coloration on the head and back above lateral line and before dorsal fin. Partially scaled.
Dominick	M	55–60 cm	Deep yellow-orange butterfly koi with dark scale outlines. Mass on left side of abdomen, left eye is disfigured and nonfunctional. Fully scaled.
Gabriel	M	55–60 cm	Yellow-gold body tapering to a deeper shade of squash towards the face with white tipped pectoral and dorsal fins. Partially scaled.
Maggie	F	50–55 cm	White with orange markings on either side of the dorsal fin above the lateral line and anterior to the dorsal fin across the medial line. Fully scaled.
Margaret	F	50–55 cm	Primarily orange and black above lateral line and white below. Partial piebald. Mouth is disfigured but functional. Fully scaled.
Gingko	M	45–50 cm	Primarily light grey-blue with dark crescent-shaped markings along the medial line and dark freckles across body and fins. Partially scaled.
Tigerlily	M	50–55 cm	Primarily orange above lateral line and white underbelly. Black markings on body and fins. Orange head and operculum with white mouth. Partially scaled.

**Table 2 animals-11-00706-t002:** Behavioral categories and definitions. Location codes calculated using SketchUp 3D modeling software and used to standardize variable zone size.

Location	Definition	Mean Area	% Total Area
Within Reach	Area of the tank designated by the arc created by the researcher using her submerged arms full range of motion in all directions without leaning forward.	0.4 m^2^	8%
Periphery	Periphery began at the boundary of within reach and extending approximately one arm’s length beyond the within reach area in all directions. In this location, the researcher’s hand was likely still within sight of the fish while also remaining out of immediate reach.	1.15 m^2^	22%
Outer Area	Included all areas of the tank outside of the previously defined locations (including areas that are out of sight in video data, see Figure 2).	3.1 m^2^	70%
Physical Interaction	Any form of tactile interaction between human hand and fish, initiated by the fish (mouthing, brushing, bumping, etc.) A series of tactile interactions were designated as a single period of interaction as long as the fish instigated physical interaction at least once over the course of 3 s and did not move away from the researcher. If interactions occurred more than 3 s apart and/or the fish moved away from the researcher between interactions, they were coded as individual periods.	N/A	N/A

**Table 3 animals-11-00706-t003:** Rate of physical interaction for each fish: overall mean rate, and by session-type (T1 vs. T2).

Rate of Physical Interaction per Minute			
Fish	Overall Rate (Mean Times/Min)	Rate by Session Type (Mean Times/Min)	
Bessie	0.02	T1	0.03
		T2	0.01
		Diff	−0.02
Dominick	0.16	T1	0.17
		T2	0.14
		Diff	−0.03
Gabriel	0.48	T1	0.63
		T2	0.29
		Diff	−0.35
Gingko	0.05	T1	0.06
		T2	0.04
		Diff	−0.02
Maggie	0.18	T1	0.24
		T2	0.10
		Diff	−0.14
Margaret	0.14	T1	0.19
		T2	0.19
		Diff	−0.1
Tigerlily	0.05	T1	0.06
		T2	0.03
		Diff	−0.02

**Table 4 animals-11-00706-t004:** Cumulative duration of physical interaction per session for each fish: overall median, and by session-type (T1 vs. T2).

Cumulative Physical Interaction Within a Single Session (20 Min)			
Fish	Duration Overall (Median and Range in Seconds)	Duration by Session Type (Median and Range in Seconds)	
Bessie	0 (0, 15.2)	T1	0 (0, 15.2)
		T2	0 (0, 6.14)
Dominick	4.4 (0, 75.9)	T1	3.75 (0.75, 7.62)
		T2	6.39 (0, 75.9)
Gabriel	37.3 (0, 281)	T1	49.1 (10.2, 281)
		T2	18 (0, 54.7)
Gingko	0 (0, 12)	T1	0 (0, 12)
		T2	0 (0, 7.68)
Maggie	11.0 (0.34, 84.7)	T1	22.8 (7.75, 84.7)
		T2	6.39 (0.34, 16.5)
Margaret	5.96 (0, 26.1)	T1	9.66 (1.27, 26.1)
		T2	3.65 (0, 11.7)
Tigerlily	0 (0, 35.4)	T1	0 (0, 35.4)
		T2	0.72 (0, 7.68)

## Data Availability

The data presented in this study are openly available in FigShare at https://doi.org/10.6084/m9.figshare.14099459.v1 (accessed on 23 February 2021).

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
