# Peer review of "Koi (Cyprinus rubrofuscus) Seek Out Tactile Interaction with Humans: General Patterns and Individual Differences"

_animals, 2021, doi:10.3390/ani11030706_

Round 1
Reviewer 1 Report
This paper is a clear, attractive and scientifically sound paper about an important and neglected subject : fish behaviour and interaction with human beings. Although I am not competent to judge the statistical details, methodology is simple, clear and well explained. Results are clearly presented, with very attractive figures. O appreciate the individual presentation of each fish. The discussion is thorough and focused. Congratulations for this interesting and good research. There is only one minor correction to be made : line 52 :
study on the implication of companion animal ownership on the heath is different from Animal Assisted Interventions, which are mostly concerned with interventions in a specific setting with a specific animal who is not owned by the person who benefits from the intervention
Author Response
This paper is a clear, attractive and scientifically sound paper about an important and neglected subject : fish behaviour and interaction with human beings. Although I am not competent to judge the statistical details, methodology is simple, clear and well explained. Results are clearly presented, with very attractive figures. I appreciate the individual presentation of each fish. The discussion is thorough and focused. Congratulations for this interesting and good research.
Thank you very much for your kind words and encouragement. We share your enthusiasm for the research and its potential.
There is only one minor correction to be made : line 52 : study on the implication of companion animal ownership on the heath is different from Animal Assisted Interventions, which are mostly concerned with interventions in a specific setting with a specific animal who is not owned by the person who benefits from the intervention.
Thank you for catching this mistake. We have corrected the sentence on lines 52-55 discussing Animal Assisted Interventions as follows:
“Historically, HAI research has focused primarily on the relationship between companion animal ownership and human health, the therapeutic effects of structured human-animal interactions (e.g. Animal Assisted Interventions), and the consequences of poor stockmanship on farmed animal production [1]” - (51-55)
Reviewer 2 Report
It is rare that I have reviewed a manuscript without finding something to comment upon. The present manuscript, however, is an exception. This paper, as the authors attest, is unique in its focus on human-fish interactions and their potential for positive fish welfare. The study is well-done, its presentation well-written, and I enjoyed reading it. I found many interesting observations that will be of use to me in my role as IACUC chair at an institution that makes great use of zebrafish. And I think the manuscript will make a great contribution to the literature. Overall, well done!
Author Response
It is rare that I have reviewed a manuscript without finding something to comment upon. The present manuscript, however, is an exception. This paper, as the authors attest, is unique in its focus on human-fish interactions and their potential for positive fish welfare. The study is well-done, its presentation well-written, and I enjoyed reading it.
Thank you! We appreciate your kind words very much.
I found many interesting observations that will be of use to me in my role as IACUC chair at an institution that makes great use of zebrafish. And I think the manuscript will make a great contribution to the literature. Overall, well done!
This is fantastic to hear! We are excited that our work will have an impact on the lives of zebrafish under your institution’s care. We appreciate your comments and thank you for taking the time to review our manuscript.
Reviewer 3 Report
The nature of human-fish interactions is an important one and the species you chose (koi) would seem particularly suited to this sort of work. However, your present experimental design fails to address any of the questions you pose. It needs, at the least, a proper set of controls before any conclusions can be attempted as to the existence of emotional interactions between koi and a'familiar' human as distinct from curiosity in the presence of a novel object, Controls should include at least an unfamiliar human hand and an artificial hand of similar shape, texture and temperature to a human hand but unconnected to a human 'maintaining eye contact' (really?).
This might permit some distinction between emotional attachment and simple curiosity. Moreover you cannot assume that fish in the absence of a novel or interesting object will experience boredom, This is a dangerously anthropomorphic assumption that ignores the possibility of 'out of sight, out of mind'. It would need, at least evidence of 'rebound' behaviour when the stimulus was returned after a period of absence.
You also do not state whether you responded to tactile contact from the fish with any movement or kept your hand still. A new experiment could compare fish responses to no movement or very gentle 'non-threatening ' movement.
I encourage you to persist with this work. However, this paper will not do.
Author Response
The nature of human-fish interactions is an important one and the species you chose (koi) would seem particularly suited to this sort of work. However, your present experimental design fails to address any of the questions you pose. It needs, at the least, a proper set of controls before any conclusions can be attempted as to the existence of emotional interactions between koi and a 'familiar' human as distinct from curiosity in the presence of a novel object, Controls should include at least an unfamiliar human hand and an artificial hand of similar shape, texture and temperature to a human hand but unconnected to a human 'maintaining eye contact' (really?).
Thank you for this encouraging evaluation of the importance of our work.
We agree that much more work is needed in this area and that controls of “an unfamiliar human hand” and/or “an artificial hand of similar shape, texture, and temperature to human hand but unconnected to a human” (as you suggest) would be particularly illuminating. Indeed, we have several similar study-designs in development and hope to be able to implement them soon. Such studies would be useful for further understanding the fishes’ interest in the experimenter and for distinguishing the source of that interest from alternative explanations such as curiosity or neophilia (interest in novelty).
For the present work, however, we had a more limited aim: simply to establish the possibility for human-fish interaction (i.e. might fish voluntarily approach and engage with a human?) and thus, whether such future studies would be of value. Given the complete absence of human-fish interactions in the published literature, the alternative possibility, that koi were uninterested in humans and/or fearful or avoidant of them (e.g., diving to the bottom of the tank, avoiding the area in which the human was situated as determined by a random motion analysis, etc.), could be considered equally if not more likely outcomes than evidence that they sought out human interaction. Put another way, the aim and value of the present work was the more modest goal of determining whether there was any phenomenon (voluntary, tactile interaction with a human) requiring further study. Our methodology and results clearly establish the presence of such a phenomenon, thus setting the mandate to conduct further follow-up studies including the ones you suggest here.
To address the important concerns you raise and to further clarify the nature of our work, study goals, and need for future investigation, we have made the following modifications (highlights indicate modified/added text).
- “By demonstrating that koi will voluntarily interact with humans and that individual differences play an important role in interaction style, this study provides the first evidence that individuated human-fish relationships may be possible, which has powerful implications for how we think about, treat, protect, and provide care for fish.” (14-17)
- “The first step in determining what role HAIs may play in fish welfare and protection is establishing whether voluntary human-fish interactions are possible. We hypothesized that the possibility for positive human-fish interactions would be evidenced by fish voluntarily approaching and interacting with the human experimenter. ” (116-120)
- “We were also interested in determining whether individual differences (i.e., personalities) played an important role in driving human-fish interactions, in which case, we expected to find stable patterns of interaction style across time.” (121-124)
- We have removed the reference to eye-contact. (186-188)
- “The following sections present a number of hypotheses regarding the findings of this study and discuss various promising avenues for future research on human-fish interactions” – (364-366)
This might permit some distinction between emotional attachment and simple curiosity. Moreover you cannot assume that fish in the absence of a novel or interesting object will experience boredom, This is a dangerously anthropomorphic assumption that ignores the possibility of 'out of sight, out of mind'. It would need, at least evidence of 'rebound' behaviour when the stimulus was returned after a period of absence.
Thank you for these observations. As you rightfully point out, we cannot provide evidence for emotional attachment to the researcher, but rather can only demonstrate that koi willingly interact with the researcher. As such, our findings may be indicative of a positive association and/or social relationship between the fish and the researcher, but further work is needed to establish this possibility. In the paper, we discuss the possibility that their behavior may also be motivated by curiosity and, as you point out, we cannot claim that our study supports any one of these hypotheses in particular due to lack of controls. We did not intend to make any definite claims but rather provide the groundwork needed to propel future investigation into these areas of inquiry. We have made the above-described modifications to indicate the restricted nature of our study goals.
Similarly, we included the discussion of boredom as it is now recognized as an important welfare consideration, especially for animals living in barren environments. As you note, our findings cannot support a conclusion that these particular fish are in fact bored. We have instead, included the reference to boredom in the hopes that researchers might use this study as a starting place for future work investigating the impacts of human interaction on the psychological welfare, including boredom, of captive fishes. We have added the following sentence to our discussion of boredom to clarify our intentions as follows:
“Future work is needed to explore this possibility.” (445-446)
You also do not state whether you responded to tactile contact from the fish with any movement or kept your hand still. A new experiment could compare fish responses to no movement or very gentle 'non-threatening ' movement.
Thank you for these suggestions. We agree that an experiment comparing fish interaction preferences (and whether the fish show individual differences in preferences for interaction-style) would be a really interesting follow-up study. To clarify the nature of our approach for the present study, we have edited the manuscript to include a description of the researcher’s physical responses to the fish on lines 185-188:
“If fish instigated physical interaction (e.g. touching, mouthing, etc.) the researchers returning contact (e.g. gently stroking the fish’s forehead, wiggling fingers, etc.)” - (186-188)
I encourage you to persist with this work. However, this paper will not do.
Thank you again for your encouragement and recognition of the importance of this work. In addressing your comments, we have clarified the nature of our study goals, how our methods address those aims, why the results matter, what future work is needed, and the limitations to the interpretation of the present work. We are very grateful for the time and thought you have devoted to our work, which is now improved as a consequence.
Reviewer 4 Report
The authors present a study wherein they have evaluated human-fish interactions with the purpose of understanding the importance of tactile pleasure/learning in captivity. Koi (correctly called Nishikigoi) as the study species is understandable because of their commercial importance. The welfare of fish is certainly understudied when compared to other vertebrates such as mammals, especially primates, and birds. This is especially evident in fish farms where I have evidenced complete disregard for their welfare and mental health displaying a complete lack of understanding of their cognitive capabilities and individual personalities.
This is a well-written paper and the flow is good. The methodology well thought out and implemented, and the statistics appropriate. The references are adequate and up to date. Although I think that a sample size of six is a bit small the arguments forwarded are convincing and a good start for bringing the subject to the attention of those who raise fish, especially for research purposes.
The only suggestion I have is that all Latin names should be italicized.
Author Response
The authors present a study wherein they have evaluated human-fish interactions with the purpose of understanding the importance of tactile pleasure/learning in captivity.
Thank you!
Koi (correctly called Nishikigoi) as the study species is understandable because of their commercial importance.
Thank you for noting the correct terminology; we have edited the manuscript to include it on lines 86-87.
The welfare of fish is certainly understudied when compared to other vertebrates such as mammals, especially primates, and birds. This is especially evident in fish farms where I have evidenced complete disregard for their welfare and mental health displaying a complete lack of understanding of their cognitive capabilities and individual personalities.
We very much agree and are encouraged that you share a similar sentiment about the current state of cultured fish welfare.
This is a well-written paper and the flow is good. The methodology well thought out and implemented, and the statistics appropriate. The references are adequate and up to date. Although I think that a sample size of six is a bit small the arguments forwarded are convincing and a good start for bringing the subject to the attention of those who raise fish, especially for research purposes.
Thank you. We agree that our sample size is somewhat small, but as you point out, we also believe that is a good starting place for bringing attention to the subject.
The only suggestion I have is that all Latin names should be italicized.
Thank you for noticing this error. We have now italicized all Latin names in the updated manuscript.
Round 2
Reviewer 3 Report
I note that you have acknowledged my criticism of the original manuscript. However, the study remains the same; it lacks proper controls so cannot be used to justify any firm conclusions as to the behaviour of koi carp. You should not seek publication until you have a properly controlled experiment to report. Then it could be very interesting.